# Array processing in cryoseismology: A comparison to network-based approaches at an Antarctic ice stream

Thomas S. Hudson[1], Alex M. Brisbourne[2], Sofia-Katerina Kufner[3], J-Michael Kendall[1], and Andy M. Smith[2]

[1]Department of Earth Sciences, University of Oxford, 3 South Parks Rd, Oxford, OX1 3AN, UK
[2]UKRI NERC British Antarctic Survey, High Cross, Madingley Rd, Cambridge CB3 0ET, UK
[3]Geophysical Institude, Karlsruhe Institute of Technology, 76131 Karlsruhe, Germany

**Correspondence:** Thomas S. Hudson (thomas.hudson@earth.ox.ac.uk)

**Abstract.**

Seismicity at glaciers, ice sheets and ice shelves provides observational constraint of a number of glaciological processes. Detecting and locating this seismicity, specifically icequakes, is a necessary first step in studying processes such as basal slip, crevassing, imaging ice fabric, and iceberg calving, for example. Most glacier deployments to date use conventional seismic networks, comprised of seismometers distributed over the entire area of interest. However, smaller aperture seismic arrays can also be used, which are typically sensitive to seismicity distal from the array footprint and require a smaller number of instruments. Here, we investigate the potential of arrays and array-processing methods to detect and locate subsurface microseismicity at glaciers, benchmarking performance against conventional seismic network-based methods for an example from at an Antarctic ice stream. We also provide an array-processing recipe for body-wave cryoseismology applications. Results from an array and network deployed at Rutford Ice Stream, Antarctica, show that arrays and networks both have strengths and weaknesses. Arrays can detect icequakes from further distances whereas networks outperform arrays for more comprehensive studies of a particular process due to greater hypocentral constraint within the network extent. We also gain new insights into seismic behaviour at Rutford Ice Stream. The array detects basal icequakes in what was previously interpreted to be an aseismic region of the bed, as well as new icequake observations downstream and at the ice stream shear-margins, where it would be challenging to deploy instruments. Finally, we make some practical recommendations for future array deployments at glaciers.

## 1 Introduction

Cryoseismology is a rapidly emerging field that shows promise for studying glaciological processes (Podolskiy and Walter, 2016; Aster and Winberry, 2017). For example, icequakes associated with basal slip (Smith et al., 2015; Roeoesli et al., 2016; Kufner et al., 2021) can provide observational constraint of frictional processes in ice dynamics models (Gräff and Walter, 2021; Hudson et al., 2023, 2020; Köpfli et al., 2022; Lipovsky et al., 2019; Zoet et al., 2013). However, glaciers are often challenging to access, logistically expensive to operate within and potential seismically active areas are typically highly variable temporally and spatially. To date, detecting subsurface icequakes has generally been performed using conventional, network-

based detection and location methods. Conventional seismic networks typically require receiver spacings similar to the distance of the events from the receiver and are predominantly sensitive to icequakes within the spatial extent of the network. Therefore, adequately sampling a specific region may require tens to hundreds of receivers. Conversely, seismic arrays are predominantly sensitive to icequakes outside the array aperture, with individual arrays requiring significantly fewer instruments than a conventional network. Seismic arrays could therefore facilitate smaller seismic deployments than conventional networks, while enabling event detection at greater distances, in a similar way to the gain provided by arrays for nuclear test ban monitoring (Bowers and Selby, 2009), for example. Here, we address the question: to what extent are array deployments useful for subsurface microseismic icequake studies. The scope of this work is to investigate the advantages, challenges and practical application of array-processing methods for icequake cryoseismology studies. Specifically, we assess the sensitivity of arrays compared to networks for: icequake detection; icequake location; whether arrays can detect event types typically missed by standard network-based detection algorithms; and discuss the limitations of arrays compared to networks. We also present an implementation of a frequency-domain array-processing method that is made available to the community to accelerate the application of array-processing in cryoseismology.

Array processing has existed as a method for decades. The core component of array processing, beamforming, involves combining plane-wave arrivals from all receivers in an array to find the direction of the origin of the wave from the array (Rost and Thomas, 2002). In this study, we focus on plane-wave beamforming, rather than on other similar methods such as matched field processing (Sergeant et al., 2020; Nanni et al., 2022), which are generally also a viable alternative to traditional network-based detection and location approaches but are beyond the scope of this study. Beamforming was originally developed for nuclear test ban monitoring (Bowers and Selby, 2009), but has since been applied to other topics, including: studying the structure of the earth (Wolf et al., 2023; Wang and Vidale, 2022; Thomas et al., 2002); monitoring offshore seismicity (Jerkins et al., 2023); ambient seismic noise source analysis (Bowden et al., 2021; Löer et al., 2018); and detection of seismicity using Distributed Acoustic Sensing instrumentation (Van Den Ende and Ampuero, 2021; Näsholm et al., 2022; Klaasen et al., 2021; Lellouch et al., 2020; Hudson et al., 2021). However, the use of array processing within cryoseismology is limited. Beamforming has been used to locate large slip events at Whillans Ice Stream, Antarctica (Pratt et al., 2014). Multiple regional arrays have been used to locate glacier earthquakes, likely caused by significant calving events (Ekström et al., 2003; Ekström, 2006; Tsai and Ekström, 2007), and studying calving processes locally (Köhler et al., 2015, 2016, 2019, 2022; Podolskiy et al., 2017). Array processing has also proven useful for locating glacier tremor (Lindner et al., 2020; Umlauft et al., 2021; McBrearty et al., 2020), near-surface icequakes associated with crevassing (Lindner et al., 2019), locating seismic sources on ice shelves (Hammer et al., 2015), and even measuring the thickness of sea-ice (Serripierri et al., 2022). The closest applications to that investigated in this study, specifically targeted at basal microseismic icequake detection, are Cooley et al. (2019) who using beamforming event location but not detection, and Lindner et al. (2019) who detect and locate surface crevassing using beamforming. Here, we present results of icequake detection and location solely using array processing and compare it to a conventional network-based approach, using a dataset from Rutford Ice Stream (RIS), Antarctica (see Figure 1). The array processing detection uses an array of 10 sensors shown in Figure 1d, with the network-based comparison performed using a network of 16 sensors (see Figure 1).

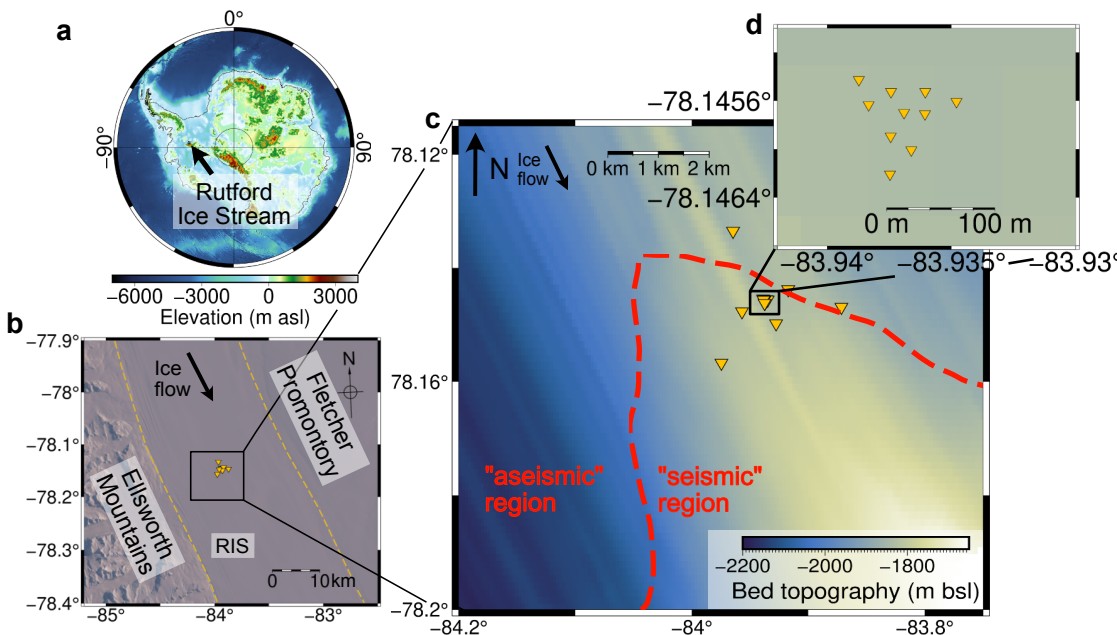

**Figure 1.** The seismic array deployment. a) Location of Rutford Ice Stream (RIS) with respect to Antarctica (basal topography is from Fretwell et al. (2013)). b) Overview of the deployment within RIS. c) Plot of the seismic network and array, as well as the previously assumed seismic and aseismic regions of the bed (Smith et al., 2015). d) Plot of array in detail.

## 2   Data

The dataset used in this study is from receivers deployed on the surface of RIS, Antarctica. The instruments deployed consist of sixteen 4.5 Hz geophones connected to Reftek RT130 dataloggers sampling at 1000 Hz, in the geometry shown in Figure 1. Ten of these instruments were deployed in a $90\ m$ aperture array, buried under several metres of snow. They were deployed in the austral summer of 2019 to 2020 (20th December 2019 to 23rd January 2020). RIS flows at $\sim 400\ m\ yr^{-1}$, with the bed known to be seismically active from previous studies (Kufner et al., 2021; Smith et al., 2015; Smith, 2006; Hudson et al.,

2021). RIS is therefore an ideal site with which to test array-based methods at glaciers.

## 3   Methods

### 3.1   Array processing

For the purposes of this study, we define a seismic array as a collection of seismic receivers configured in a geometry that allows for the data to be processed using beamforming methods. In comparison, we define a network as a collection of seismic

receivers that are used to perform conventional seismic detection and location, spaced so as to be most sensitive to events inside the network.

### 3.1.1 Detection and location

There are several different methods for detecting and locating icequakes using arrays. Here, we describe the specific method chosen for this study, a frequency-domain, frequency-wavenumber (fk) beamforming method, chosen because it is more computationally efficient than time-domain methods (Rost and Thomas, 2002). Similar methods are routinely deployed by various seismic observatories globally (Schweitzer et al., 2009).

The method comprises of four overall steps (see Figure 2):

1. Compute the beam-power, slowness and back-azimuth time-series (*P(t), S(t),* Θ*(t)*).

2. Detect and associate icequake phase arrivals.

3. Locate each icequake (via either full-3D or fixed-depth methods).

4. Apply earthquake slowness and beam-power filters in an attempt to remove falsely triggered events.

*P(t), S(t)* and Θ*(t)* are calculated using the frequency-wavenumber method as follows. First, the discrete frequencies to perform beamforming on are specified. Here, we use 20 values between *10 Hz* and *150 Hz*, the bandwidth within which icequakes have previously been observed at RIS (Smith et al., 2015; Kufner et al., 2021). The slowness limits for the beamforming calculation are then defined. The maximum slowness for this study is defined as *1 s $km^{-1}$*, which is chosen based on an extreme estimate of the minimum apparent velocity (*1 km $s^{-1}$*) that one might expect to detect for an icequake at RIS. We also make the assumption of a plane wave incident at the array. This is approximately valid for a *90 m* aperture array with a minimum hypocentral distance of *2.2 km* (ice thickness). The velocity time-series data at each receiver are then windowed in time, in order to calculate beam-power, slowness and back-azimuth for each time window. The time window used here is *0.2 s*. This value is greater than the slowest time it would take for a seismic wave to travel across the array and provides frequency resolution up to 5 Hz. To provide sufficient phase-arrival time resolution the window time-step is *0.01 s*, hence consecutive windows overlap. The beam-forming is performed as follows. The total beam energy, $E(\boldsymbol{u_i}, \theta)$ for a given slowness $\boldsymbol{u_i}$ and back-azimuth $\theta_i$ over a given time-window, $t_{win}$, is calculated as follows (Rost and Thomas, 2002; Bowden et al., 2021),

$$E(\boldsymbol{u_i}, \theta_i) = \frac{1}{N} \int_{-\infty}^{-\infty} \sum_{n=1}^{N} s_n(t + \Delta t_i)^2 dt,  \tag{1}$$

where $s_n(t)^2$ is the power spectral density of the time-series at a single receiver $n$ and $\Delta t_i$ is the time-shift required to align receiver $n$ with the centre of the array,

$$\Delta t_i, n(\boldsymbol{u_i}, \theta_i) = \begin{pmatrix} u_{x,i} sin(\theta_i) \\ u_{y,i} cos(\theta_i) \end{pmatrix} . \boldsymbol{r_n},  \tag{2}$$

for an array with negligible surface topography variation and where $r_n$ is the distance of receiver $n$ to the centre of the array. Here, we undertake beamforming in the frequency-wavenumber (fk) domain, where Equation 1 can be rewritten as,

$$E(\boldsymbol{u_i}, \theta_i) = \frac{1}{N} \int_{-\infty}^{-\infty} |\sum_{n=1}^{N} S_n(\omega) e^{2\pi\omega\Delta t}|^2 dt = \frac{1}{N} \int_{-\infty}^{-\infty} |\sum_{n=1}^{N} S_n(\omega) e^{2\pi\omega\boldsymbol{u_i}\boldsymbol{r_i}}|^2 dt, \tag{3}$$

where $S_n(\omega)^2$ is the power spectral density of the time-series in the Fourier domain and $\boldsymbol{k_i} = \omega\boldsymbol{u_i}$, is the wavenumber associated with the given slowness and angular frequency. Here, we linearly stack over all discrete frequencies, so the integral in Equation 3 simply becomes a linear sum. The power for a particular slowness and backazimuth, over a particular time-window, $P(\boldsymbol{u_i}, \theta_i, t)$ is hence given by,

$$P(\boldsymbol{u_i}, \theta_i) = \frac{E(\boldsymbol{u_i}, \theta_i)}{t_{win}}. \tag{4}$$

One now has the beam-power for all points in the 2D slowness space, for each time window (see Figure 2c). From this 2D slowness space, the peak beam-power $P(t)$ and its associated slowness $S(t)$ and back-azimuth $\Theta(t)$ can be calculated for each window. This process is performed for the vertical and both horizontal components of the seismic data separately.

$P(t)$, $S(t)$ and $\Theta(t)$ can then be used to detect icequakes. For this study, the vertical component beam-power time-series, $P_V(t)$, is used to detect potential P-wave arrivals and the two horizontal beam-power time-series are summed to give $P_H(t)$, used to detect potential S-wave arrivals. Potential P- and S- wave arrivals are detected by searching for peaks in $P_V(t)$ and $P_H(t)$, respectively (see Figure 2d). Here we assume that a steep velocity gradient due to a near-surface firn layer causes approximately all P-wave energy to be incident on the vertical component and approximately all S-wave energy to be incident on the horizontal components. Surface waves may have energy on both vertical and horizontal components too, but such phases are hopefully removed from the catalogue by the slowness-ratio filter. Peaks in $P_V(t)$ and $P_H(t)$ are identified using an median absolute deviation (MAD) threshold, with any peak in $P(t)$ with a MAD multiplier value of 2 (optimised for this dataset) and a minimum event time separation $> 0.25$ $s$ defined as a potential phase arrival (based on the minimum P-S separation time for a basal icequake). Potential P- and S- phases then need to be associated. We only trigger an icequake detection if we can associate both P-wave and S-wave arrivals for an event. The phase association algorithm is as follows, with an example of the result shown in Figure 3a. First, only potential P and S arrivals with a maximum separation of $\Delta t_{P-S,max} = 10s$ are considered, to minimise mis-associating incorrect arrivals. The value of $\Delta t_{P-S}$ used limits the hypocentral distance to which we can detect earthquakes (see Equation 6). In order to account for possible overlapping event arrivals, only P and S arrivals with back-azimuths in agreement within $15^o$ are paired. A third criterion is that if multiple P and S arrivals meet the first two criteria, then the highest beam-power P arrival is associated with the highest beam-power S arrival that has a back-azimuth difference $< 15^o$ and a time difference $< \Delta t_{P-S}$, and so on for consecutively lower beam-power arrivals. In practice, the maximum-beam-power criterion is rarely required, as two events are normally distinguishable within the $\Delta t_{P-S} = 10$ $s$ window by back-azimuth alone.

The phase-associated icequake arrivals can then be used to locate events. Array-based icequake location differs from network-based methods in that instead of many P- and S- phase arrivals used to locate an event, one instead has single P

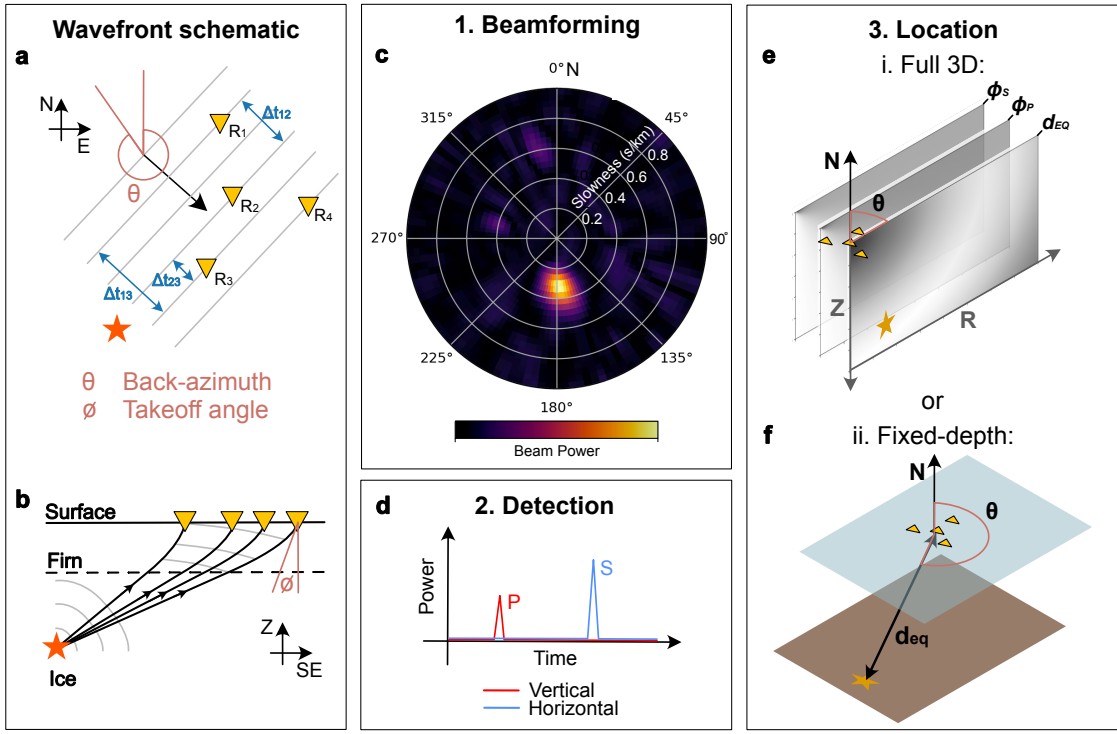

**Figure 2.** a,b. Plane wave arriving at array in the horizontal and vertical planes, respectively. All arrivals are time-shifted to align arrivals. (diagram inspired by Bowden et al. (2021)) c. Heatmap of beam-power in slowness-back-azimuth space for one time window. An event is shown by the peak in beam-power. d. Schematic example of power time-series, with vertical peaks associated with P arrivals and horizontal peaks with S arrivals. e. Array-based 3D location algorithm, involving stacking PDFs of P-S travel-time, P-wave takeoff angle and S-wave takeoff angle. f. Fixed depth array-based location using back-azimuth and P-S travel-time only.

and S arrival observations, but with the additional information of slowness and back-azimuth associated with these two phase arrivals. To use an array to locate an icequake in 3D, one has to calculate a takeoff angle, $\phi_{EQ}$, and a radial distance, $d_{EQ}$, from the event to the receiver, in addition to using the back-azimuth. In this study, we compare two methods of finding ice-quake hypocentres: a 3D location method and a fixed-depth method (see Figure 2e,f). The motivations for applying these two methods are discussed in Section 4.2. The takeoff angle for phase $i$ (P or S), $\phi_{EQ,i}$, is calculated from the apparent velocity and slowness, $v_{app,i} = 1/S_{app,i}$, and the approximate seismic velocity at the receiver, $v_{rec,i}$, using the equation,

$$\phi_{EQ,i} = sin^{-1}\left(\frac{v_{app,i}}{v_{rec,i}}\right).$$

(5)

In both cases, $d_{EQ}$ is calculated using the P-S travel-time delay, $\Delta t_{P-S}$, using the equation,

$$d_{EQ} = \frac{v_P v_S}{v_P - v_S}\Delta t_{P-S},$$

(6)

where $v_P$ and $v_S$ are the path-average P- and S-wave velocities, respectively. These are approximated here to be the bulk ice velocities ($v_P = 3841 \ m \ s^{-1}$, $v_S = 1970 \ m \ s^{-1}$), obtained using a relationship between seismic velocity and temperature (Smith, 1997; Smith et al., 2015; Kohnen, 1974). In the 3D location method, for a vertically homogeneous ice structure, the raypath from icequake source to receiver is linear, so the takeoff angle, back-azimuth and hypocentral distance ($d_{EQ}$) can be used to directly back-project the event location in 3D. However, at RIS there is a $\sim 100 \ m$ thick firn layer directly below the surface, which has increasingly slow seismic velocities towards the surface (Smith et al., 2020; Zhou et al., 2022) (see Section 3.1.2 for details). This causes the rays to dip steeply vertical as they near the surface. This effect has to be accounted for when locating icequakes, since the ray path cannot be assumed linear near the surface, with the takeoff angle representing the final ray path angle rather than the initial takeoff angle at the source. To account for a firn layer, we specify a vertically varying but laterally homogeneous velocity model and perform ray-tracing to every grid point on a 2D depth/horizontal-radial-distance grid from the array (at *10 m* grid resolution), to create a 2D spatial takeoff angle Probability Density Function (PDF) (see Figure 2e). Similarly, we create a second 2D spatial PDF for $d_{EQ}$, using $\Delta t_{P-S}$ for a given icequake (see Equation 6) and assuming an appropriate uncertainty, $\delta d_{EQ}$, again in depth and horizontal distance from the array, for $d_{EQ}$. The 2D takeoff angle and $d_{EQ}$ PDFs are then stacked and normalised to create an overall misfit space (see Figure 2e), where the most likely icequake location within the 2D plane corresponds to the peak in this combined PDF space (see Figure 2e). The 3D icequake location is then calculated by including the back-azimuth information. Conversely, in the fixed depth method, the icequake location is calculated without using the takeoff-angle, instead simply projecting the distance $d_{EQ}$ onto a fixed-depth horizontal plane at the measured back-azimuth (see Figure 2f). The fixed depth horizontal plane is equal to the ice thickness, explicitly assuming that all events originate at the bed.

The initial icequake catalogue will likely contain both real events and false triggers. In order to discriminate between real events and false triggers, we apply two filters. The first filter is a slowness-ratio filter. In this work we are interested in subsurface icequakes, where the array-processing has correctly identified P- and S- body-wave phases. P-, S- and surface wave phases all have different propagation velocities and so will have different slownesses (since slowness is the inverse of velocity). However, the beamforming algorithm used here calculates apparent slowness rather than actual slowness, which is dependent upon the angle of incidence of the wave with respect to the surface. Apparent slowness cannot therefore be used to discriminate between phase arrivals. Instead, we use the slowness ratio ($S_S/S_P = v_P/v_S$), which is independent of apparent slowness effects if both the P- and S- waves have approximately the same ray-path. A histogram of $v_P/v_S(= S_S/S_P)$ ratio for the initial icequake catalogue is plotted in Figure 4a. Ice at RIS has a $v_P/v_S$ ratio of 1.95 (Smith et al., 2015), approximately corresponding to the peak in Figure 4a. We remove any events with $S_S/S_P < 1.8$ and $S_S/S_P > 2.1$. This filters our catalogue from $\sim 700,000$ events to $\sim 250,000$ events. The second filter we apply is a beam-power filter, only keeping events with a combined beam-power from both the P and S arrivals greater than $2 \times 10^8 \ counts^2$. This filter will likely remove some real events as well as false triggers, since smaller events that are further away from the array will inevitably have smaller beam-power phase arrivals. Although the beam-power filter is likely less sensitive to whether an event is real or a false trigger than the slowness filter, we still apply it in an attempt to remove more false triggers, so as to obtain a catalogue with hopefully a higher real event to false trigger detection rate. We use a beam-power filter of $2 \times 10^8 \ counts^2$ based on the assumption that englacial events directly

beneath the array (with depths $< \sim 2\ km$) at RIS are likely false triggers (see Figure 4b). The beam-power filter reduces our catalogue from $\sim 250,000$ events to $\sim 25,000$ events, which is herein referred to as the array-based icequake catalogue.

### 3.1.2 Velocity model

At Rutford Ice Stream, there is a $\sim 100\ m$ thick firn layer, overlying an assumed isotropic bulk ice column of several km thickness. Bulk ice velocities are assumed to be $v_P = 3841\ m\ s^{-1}$, $v_S = 1970\ m\ s^{-1}$, for the P and S waves, respectively. These are obtained using a relationship between seismic velocity and temperature, calibrated using seismic refraction methods (Smith, 1997; Smith et al., 2015; Kohnen, 1974). The 100 metre firn layer velocity model used in this paper is based on that derived by refraction seismic methods (Smith et al., 2020), which has been approximately independently verified using ambient seismic noise methods (Zhou et al., 2022). The full velocity model is included in a data repository (see Code and data availability for details). The velocity model used for 3D location increases vertically from $v_P = 660\ m\ s^{-1}$, $v_S = 338\ m\ s^{-1}$ at the surface approximately exponentially to $v_P = 3841\ m\ s^{-1}$, $v_S = 1970\ m\ s^{-1}$ at $110\ m$ below the surface and beyond, as in Smith et al. (2020). For the fixed depth method, a homogeneous velocity model using the bulk ice velocities above is used. We assume that uncertainty in the velocity model are negligible compared to uncertainty in phase arrival times.

### 3.1.3 Uncertainty estimation

There are various sources of uncertainty in icequake phase arrival times and hypocentres. Sources include uncertainty in the velocity model, the GPS-derived receiver locations and potentially anisotropic effects. Here, we estimate the overall uncertainty in event phase arrival times and location as follows. Phase arrival time uncertainties are defined as the full-width half maximum of the peak in the beam-power time-series associated with the particular phase arrival. Spatial uncertainties are defined by uncertainty in slowness and back-azimuth. The uncertainty in slowness is defined as the full-width half maximum of the peak in beam-power in the radial direction of the 2D slowness space (Figure 2c). Similarly, the uncertainty in back-azimuth is the full-width half maximum of the peak in beam-power azimuthally in the 2D slowness space of Figure 2c. In each case, the full-width half maximum is used rather than the standard deviation of a Gaussian fit to improve computational efficiency. We assume that there is negligible uncertainty in the velocity model. However, in reality uncertainty in the velocity model will likely contribute to uncertainty in event locations that we have not accounted for in the array-processing derived icequake locations.

### 3.1.4 Array sensitivity

Seismic arrays are sensitive to icequakes with a particular bandwidth, which is governed by the minimum and maximum spacing between individual receivers. The minimum optimal frequency that an array is sensitive to is proportional to the maximum receiver spacing and vice versa for the maximum optimal frequency. The equation that describes this is given by (Bowden et al., 2021),

$$f = \frac{v}{x_{rec}}, \tag{7}$$

where $f$ is frequency, $v$ is the seismic velocity and $x_{rec}$ is the receiver spacing. For the array in this study, the minimum and maximum receiver spacings are *20 m* and *90 m*, respectively. The corresponding optimal bandwidth for the array is *40 Hz* to *200 Hz* for P-waves and *20 Hz* to *100 Hz* for S-waves, using the bulk ice velocity for P and S waves, respectively. Depending upon the array geometry and level of radial symmetry, the receiver spacing and therefore sensitivity could vary with azimuth. The array in this study is designed to be approximately radially and azimuthally symmetric, so the azimuthal sensitivity is approximately constant.

### 3.1.5 A note on stacking

Stacking the phase-shifted waveforms within the beamforming process suppresses incoherent noise. In this study we linearly stack the data. However, one could also use nth root stacking (Rost and Thomas, 2002) or phase-weighted stacking (Schimmel and Paulssen, 1997). One could also consider stacking different frequency ranges separately, in order to further improve sensitivity (Gal et al., 2014). We find negligible differences in performance using different stacking techniques for the array setup in this study. This is likely because the array has a small (*100 m*) aperture and minimal scattering occurs within the shallow firn layer at RIS, therefore resulting in insignificant incoherent noise compared to local icequake signals, regional earthquakes and coherent ambient seismic noise sources. We would recommend reconsidering the choice of stacking method for larger aperture arrays or sites with significant near-surface heterogeneity, either of which would increase incoherent noise levels.

### 3.2 Network-based icequake detection and location

The array-based detection and location method presented in this study is benchmarked against a current network-based migration method for icequake detection, QuakeMigrate (Hudson et al., 2019). This method approximates the energy from an icequake arriving at each receiver with a Gaussian onset function, which are then back-migrated through time and space to search for a coalescence of energy that corresponds to an event. If a sufficiently high coalescence of energy is found over a particular time window, then it triggers an event detection. Full details on this method can be found in Hudson et al. (2019). QuakeMigrate provides an estimate of hypocentral location for each event that equates to the 3D grid node corresponding to the maximum coalescence of energy. We therefore relocate all events using the non-linear, probabilistic earthquake location algorithm NonLinLoc (Lomax and Virieux, 2000), in order to obtain a precise location and physically meaningful estimates of hypocentral uncertainty. The QuakeMigrate method shares similarities with the beamforming method (see Section 3.1), with both seeking to identify coherent arrivals of energy within a particular frequency bandwidth. The fundamental difference between the two methods is that for plane-wave array processing, earthquake sources need to be sufficiently far away that the plane-wave assumption holds, whereas the network-based method is optimised for sources within the network extent. Another key difference between the two is that the array-based method searches a pre-defined slowness-azimuth space, which in this case is undertaken in the fk-domain, while the network-based approach searches over a pre-defined 3D spatial grid through time.

### 3.3 Moment magnitude

One can use the icequake magnitude distribution to quantify the sensitivity of the array compared to the network, for smaller vs. larger events. We choose to use moment magnitude, $M_w$ (Hanks and Kanamori, 1979), as it provides an absolute measure of the actual moment release of an icequake, rather than relying on an empirical relationship. Another benefit is that unlike local magnitude scales, it doesn't exhibit a break in the scaling relationship at low magnitudes ($M_w < 3$) (Deichmann, 2017; Hudson et al., 2022). Moment magnitudes are calculated using SeisSrcMoment (Hudson, 2020) (see Hudson et al. (2022) for details on applying the method). This involves fitting a Brune model (Brune, 1970) to the frequency spectrum of the icequake.

## 4 Results and discussion

### 4.1 Icequake detection using arrays vs. networks

Detected icequake hypocentres are shown in Figure 5, with a summary of the magnitude distribution of these icequakes shown in Figure 6. The magnitude distributions for three detection setups are shown in Figure 6: (1) the array-based detection method applied to the array shown in Figure 1d (Figure 6b); (2) the network-based detection method applied to the same array shown in Figure 1d (green points, Figure 6a); and (3) the network-based detection method applied to the entire network shown in Figure 1c (red points, Figure 6a). The results allow for the comparison of an array deployment to a network deployment, as well as an array-based vs. network-based detection and location algorithms more generally.

The array-based detection outperforms the network-based detection method in several areas. Firstly, the array detects more icequakes than the network (see Figure 6). The additional icequake detections in the array data originate predominantly at greater distances from the array/network centre than icequakes detected using the network-based approach (see Figure 5). Icequakes continue to be detected at distances of 10 $km$ or more, albeit with an increasing average magnitude with distance, whereas the network has a sharp detection limit at $\sim 7.5$ $km$ from the network centre (see Figure 6d). The sharp limit for the network-derived catalogue is the consequence of the boundary of the network-based search grid, with the search area significantly exceeding the physical microseismic detection limits of the network-based algorithm. Additionally, the array detects $\sim$1,000 icequakes with magnitudes $> 0$, whereas the network only detects a negligible number of these larger icequakes, many of which originate far outside the spatial extent of the network (see Figure 6). The array-based method also detects a higher proportion of the smaller icequakes, with a magnitude of completeness, $M_{c,array} = -1.4$, compared to $M_{c,network} = -0.81$ for the network-based method.

However, the network-based detection outperforms the array-based detection method for discriminating false triggers. This is evidenced by the clustering of seismicity at sticky-spots clearly exhibited in the network-based event locations (Figure 5c). The array-based data does exhibit some clustering near the array (Figure 5a), but as hypocentral distances from the array increase, the events become more scattered, likely a combination of both false-triggers and poorer hypocentral constraint.

Overall, the array-based method is more sensitive than the network-based method, detecting more icequakes across the magnitude range. This result is likely for two reasons. Firstly, the array-based method outperforms the network-based approach

for phase association, with multiple P- and S- wave phase arrivals possible to associate within a given time-window. This is possible due to the accuracy of back-azimuth measurements, allowing arrivals with back-azimuths differing by $< 15^o$ to be paired, even if phase arrivals overlap. This is particularly powerful when accounting for radiation pattern effects, as shown in Figure 3. For the example in Figure 3, two events close in time originating from different back-azimuths have inverse P/S amplitude ratios, due to radiation pattern effects, yet are both detected within the same window. Theoretically, tens of events could overlap within each time window, limited only by the back-azimuth tolerance and distribution of event back-azimuths. This is in contrast to the network-based approach, where only one event association would be allowed within a given time window, so as to minimise the risk of incorrect phase associations. However, the greater number of events detected by the array-based method could also be due to a lack of metrics to filter the catalogue by. Although our array-processing method provides uncertainty estimates, these have a particularly coarse temporal resolution, limiting their use for filtering the data to remove false event detections. Conversely, the network-based approach measures uncertainty with a higher resolution, allowing both temporal and spatial uncertainty filters to be used to remove false detections. However, given our strict ($< 15^o$) back-azimuth phase association criterion and slowness-ratio filter, we are confident that the difference between the array-based and network-based methods cannot be attributed solely to false event triggering.

### 4.1.1   A note on icequake magnitude distributions

Tectonic earthquake magnitudes typically follow a logarithmic scaling relationship (Gutenberg and Richter, 1936, 1944). The array-based icequake detection results shown in Figure 6b exhibit a similar relationship, with the tail-off at magnitudes below the magnitude of completeness, $M_c$, caused by icequake signal-to-noise (SNR) ratios falling below the noise level, leading to not all events being detected. However, the network-based icequake magnitude distributions do not exhibit a clear linear trend (see Figure 6a). This effect is not caused by S-wave anisotropy or assumptions about the source mechanism orientation, with P-wave $M_w$ and average moment-tensor $M_w$ distributions exhibiting similar peaks and troughs in the binned data. We instead find that these peaks and troughs in the $M_w$ distribution are caused by spatially-distinct icequake clusters with their own narrow magnitude distributions (see Figure 6c). The limited extent of magnitude variation for each cluster is presumably governed by bed properties, whether that be the extent of slip, the rupture velocity of the ice-bed interface, or other similar effects (Gräff and Walter, 2021; Hudson et al., 2023; Zoet et al., 2012). This clustered distribution of icequakes also likely plays a role in mean $M_w$ with distance (see Figure 6d), where the network is more sensitive to small icequakes in clusters within the network extent. The array-based detection results do not exhibit this cluster-dominated behaviour since it can detect events at greater distances, therefore sampling a greater distribution of icequake clusters and potentially icequake sources.

### 4.2   Icequake location using arrays vs. networks

The network outperforms the array for icequake location. This is evident from the clearly discernible icequake clusters in the network-based icequake catalogue shown in Figure 5c, expected at RIS based on findings from the same area of bed when a much denser seismic network was deployed (Kufner et al., 2021). Similar observations are obtained when locating the icequakes only using the ten inner array stations with the network-based location method. Icequake clustering is generally

indiscernible in the array-based icequake location results in Figure 5a. One reason for this is the filtering of high quality icequake phase arrivals in the network data using temporal and spatial uncertainty measurements, as described in Hudson et al. (2019). Noisy, poorly-constrained yet real icequakes are likely filtered out by definition in the network-based results whereas these noisier, low SNR icequakes are more likely kept in the array-based catalogue compared to the network-based catalogue. This could also affect the behaviour of $M_w$ with distance (see Figure 6d).

However, there is also a more fundamental limitation in the location results of the array-based method: the presence of a near-surface, low-velocity firn layer (Smith et al., 2020; Zhou et al., 2022) that causes seismic waves to steeply dip towards the surface (see Figure 2). Glacier settings with a thinner or no firn layer would result in better constraint of event location. Uncertainty in the velocity structure of the firn layer, especially at P-wave wavelengths ($< 10\ m$) limits the measurement of takeoff angle from apparent slowness used in the array-based method's 3D icequake location procedure. This is what causes the icequakes located using the array-based method to be miss-located, approximately directly beneath the array (red scatter points, Figure 5e,f). The firn-layer effect on the 3D location method is most pronounced in Figure 5f, where events are projected below the ice-bed interface. To mitigate this issue, we are forced to neglect firn layer effects, and project icequake epicentres onto an artificial horizontal plane at approximately the depth of the ice-bed interface using the fixed-depth location method (see Figure 2f). Obviously this is an approximation, with both neglecting the firn layer and differences between the average ice-bed interface and the true icequake depth resulting in greater uncertainty in the icequake epicentres, likely making any icequake clusters challenging to discern. High resolution ($< 10\ m$) 3D imaging (lateral in addition to vertical) of the firn velocity structure beneath the array could allow one to apply an array transfer function to minimise these effects, although without access to such data we cannot test this hypothesis. We therefore include the 3D-method location results to emphasise the importance of understanding the near-surface velocity structure when performing array processing.

## 4.3 New insights into Rutford Ice Stream

The array deployment provides new insight into seismicity at RIS. Previous studies of RIS suggest that bed properties vary upstream of the deployment vs. downstream (Smith, 1997; Smith and Murray, 2009). The upstream bed is thought to be comprised of unconsolidated sediment that fails aseismically, while the bed downstream can fail seismically at sticky-spots (Smith et al., 2015) (see Figure 1). Kufner et al. (2021) use a larger network of 35 receivers to find icequakes upstream of the seismic-aseismic boundary (blue points and purple line, respectively, Figure 7a). These are interpreted to occur in the depressions of Mega-Scale Glacier Lineations (MSGLs) in the previously aseismic region. Array-processing enables event detection at greater hypocentral distances, allowing us to confirm the findings of Kufner et al. (2021), with seismicity extending further upstream again, likely only limited by array sensitivity, the maximum $\Delta t_{P-S,max}$ time that we impose for P-S phase association, and our false-trigger beam-power filter. Figure 7b shows an example of a previously undetected upstream event. The icequake has high SNR P-wave and S-wave arrivals. The network-based method would likely fail to detect the event even if the search grid were sufficiently large, since there are multiple phase arrivals in the window from other, overlapping event. Overall, observing seismicity in the previously inferred aseismic region has implications for bed complexity and icequake

nucleation, potentially supporting ideas such as basal water pressures modulating bed friction and seismicity (Hudson et al., 2023; Gräff et al., 2021).

The array-derived icequake catalogue also contains earthquakes further downstream than found in Kufner et al. (2021) (see Figure 7a). It is expected that events might originate from this region. However, what is surprising is the number of seismic signals in the time-series (see Figure 7c). Typical icequake repeat times for individual spatial clusters are of the order of 100s to 1000s seconds at RIS (Hudson et al., 2023) and the waveforms do not look similar, so the other potential events within this window likely originate from various locations. The high number of signals within the 6 second window shown in Figure 7c would likely be challenging to separate using the network-based detection algorithm but the array-based algorithm can associate P and S phases based on the slowness measurements. This emphasises the phase-association benefits provided by the array-based method compared to the network-based detection method.

The final observation in the array results that we emphasise here are icequakes at the shear-margins of RIS. Again, these events are observed due to the greater detection distances of array-based methods (up to $20\ km$) compared to networks ($< 7.5\ km$). An example icequake from the shear-margin is shown in Figure 7d. This event also likely originates at or near the glacier bed, since the slowness-ratio false-trigger filter applied to the icequake catalogue should remove any surface wave detections. The large amplitude S-wave compared to the P-wave is likely a combination of the position on the icequake focal sphere and perhaps also the higher shear-rates near the ice stream shear margins. The S-wave also appears to potentially exhibit shear-wave splitting associated with seismic velocity anisotropy. Detecting such icequakes could provide information on shear-margin dynamics, such as how important damage to the ice fabric is for impeding or accelerating ice stream flow.

## 4.4 Lessons learnt and recommendations for using arrays in the cryosphere

1. *Seismic networks are more sensitive than arrays for studying smaller areas in more detail.* The network-based approach had a smaller magnitude of completeness than the array-based approach, while also providing greater spatial constraint of icequake hypocentres that allow clusters of events to be identified. For studying a particular glaciological process in as much detail/as comprehensively as possible, we recommend deploying a seismic network rather than an array. This is especially true for sites with a thick (greater than seismic wavelength) low-velocity firn-layer.

2. *Arrays can detect icequakes at greater distances than networks.* Arrays can outperform networks for detecting events, especially when P- and S-wave phase arrivals overlap in time. Theoretically, the array-processing method here can detect many events within a given time-window, compared to a single event using the network method. Furthermore, phases from an event that have highly differing SNRs can still be readily associated as from the same event. In this work we implement a standard fk-domain method, stacking all frequency ranges equally, but the sensitivity of our method for simultaneous event detection could be further improved by stacking different frequency ranges separately (Gal et al., 2014). Together, these properties of the array-processing method enable more events to be detected from greater distances, and during noisier time periods. We therefore suggest that arrays are useful for initial scoping of a field site, before potentially deploying a more comprehensive network to study a particular process.

3. *Arrays typically require fewer instruments than a network.* The array in this study comprised of only ten receivers, whereas ideally a seismic network would comprise of tens to hundreds or more receivers. Arrays therefore provide an efficient means to investigate seismic activity, at least initially.

4. *Array and network geometries limit detection performance in different ways.* As described above, arrays and networks have different advantages and compromises. Arrays require receiver spacings optimised to the spectral-content of the earthqaukes to be detected, which may not be known in advance. In this study that spacing is *20 m* to *90 m*. In contrast, networks generally perform best when receivers are evenly spaced, with icequake depths are not greater than the maximum receiver spacing. The optimal receiver spacing for a network to study basal icequakes at RIS ($\sim 2\ km$) is therefore

much greater than the maximum optimal array receiver spacing ($\sim 100\ m$). To summarise, it is challenging to design a deployment optimised for both array and network processing.

5. *Consider networks of sub-arrays to capitalise on the advantages of both network and arrays.* If one has a sufficient number of receivers, then it may be possible to deploy sub-arrays, evenly spaced within an overall network. This could facilitate a hybrid detection approach, taking advantage of both network and array benefits.

6. *Consider generating a more complete catalogue of events based on the initial catalogue using other methods.* Once a catalogue of initial events has been obtained, one could use other methods such as template-matching to increase the number of events detected (Helmstetter, 2022; Gimbert et al., 2021; Gibbons and Ringdal, 2006).

## 5  Conclusions

Here, we focus on how useful array-processing is for deployments used to study icequakes at glaciers. The motivations for using
arrays rather than networks are that the cryosphere is often challenging to access, logistically expensive to operate within and potential seismically active areas are typically highly variable temporally and spatially. Seismic arrays could facilitate smaller seismic deployments than conventional networks, while enabling event detection at greater distances. We find that arrays can detect icequakes over a greater spatial extent than seismic networks, but provide poorer spatial constraint on seismicity within a network, where networks have the potential to elucidate glaciological processes in greater detail. Array slowness-ratios play
an important role for discrimination of real events from false triggers at greater distances. At Rutford Ice Stream, events are detected in what was previously thought to be an aseismic region with different bed properties, downstream where icequakes have never previously been observed, and at the otherwise inaccessible ice stream shear margins. These results emphasise the value of array-based icequake detection for more comprehensively studying glacier dynamics over larger spatial footprints than otherwise possible. We suggest a number of recommendations based on learnings from this study, especially that arrays might
be particularly useful for initial scoping field deployments, where one only has access to an insufficient number of instruments to deploy a suitable network, or for regions that are potentially hazardous to operate in, such as crevassed shear-margins.

*Code and data availability.* The beamforming method presented in this study is available as an open source python package, SeisSeeker (https://zenodo.org/badge/latestdoi/523305896). The comparison network-based method, QuakeMigrate, is also available open source (Winder et al., 2021). The velocity model used and the final seismic catalogues containing all detected events, their locations and magnitudes are available here (https://doi.org/10.5281/zenodo.8120941). Continuous seismic data for the entire experiment period are available on IRIS (network code 6L, for 2019-2020, doi: https://doi.org/10.7914/SN/6L_2019).

*Author contributions.* AB and SK designed the experiment. TH developed the method, performed the analysis and prepared the original draft. AB and AS acquired the funding. All authors contributed to the review and editing of the manuscript.

*Competing interests.* The authors have no competing interests.

*Acknowledgements.* We thank C. Thomas for her advice on designing the array geometry. We thank the authors of Bowden et al. (2021) for sharing python notebooks that inspired the underlying beamforming algorithm used in this work. We highly recommend that those interested in the method also read this work and associated python notebooks. We also thank A. Köhler and an anonymous reviewer who provided valuable comments that have no doubt improved the work. We thank the UKRI NERC British Antarctic Survey for providing logistical support for the project, which was funded by the BEAMISH project (grants: NE/G014159/1, NE/G013187/1). The instruments used for this
experiment are from the UK Geophysical Equipment Facility (GEF loan number 1111). T. Hudson was funded by The Leverhulme Trust via a Leverhulme Early Career Fellowship (ECF-2022-499).

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

560

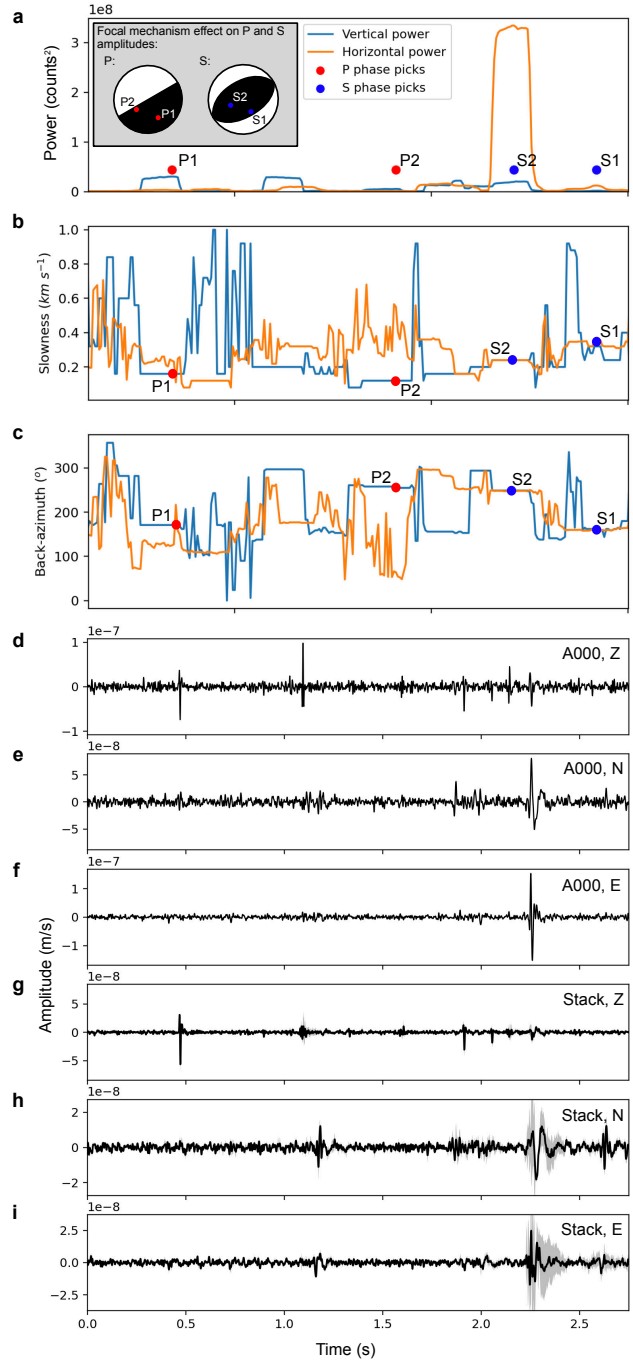

**Figure 3.** Example of array phase detection and association performance. a. Plot of beam-power and peaks associated with phase arrivals for two events. Inset diagram shows possible focal mechanisms that could cause the observed P- and S- amplitudes. d-f. Waveforms arriving at centre array station. g-i. Mean of stacked and time-shifted waveforms for the entire array, using the slowness and back-azimuth of event 2 in (b),(c).

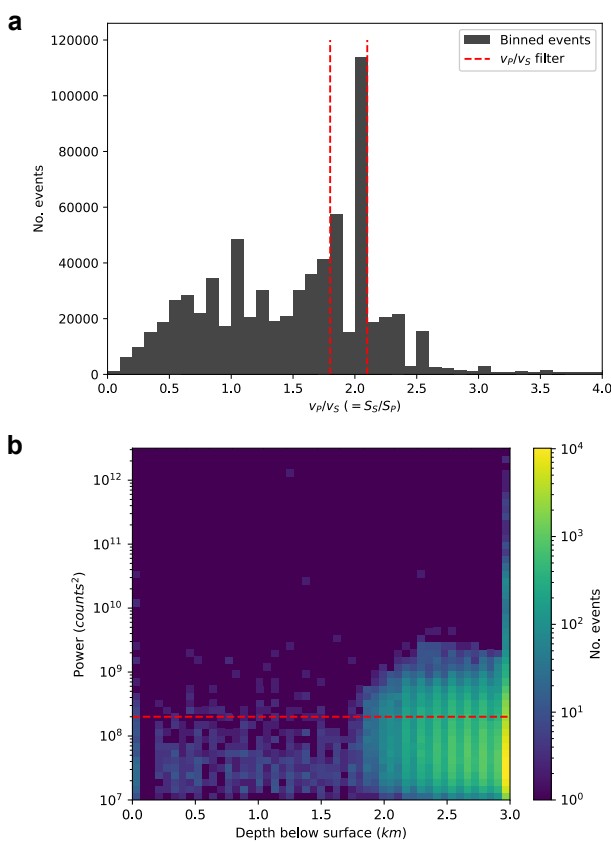

**Figure 4.** Summary of how the initial array-processing derived icequake catalogue is filtered. a. Histogram of slowness-ratio for all earthquakes in the initial catalogue (note that $v_P/v_S = S_S/S_P$). Red dashed lines indicate filter values used to produce the final icequake catalogue. b. Histogram of combined beam-power ($P_P + P_S$) vs. depth below surface for icequakes with $1.8 < v_P/v_S < 2.1$, with depth derived using the 3D location method. Ice thickness is $\sim 2.2km$. Red dashed line indicates the minimum beam-power filter used to obtain the final icequake catalogue.

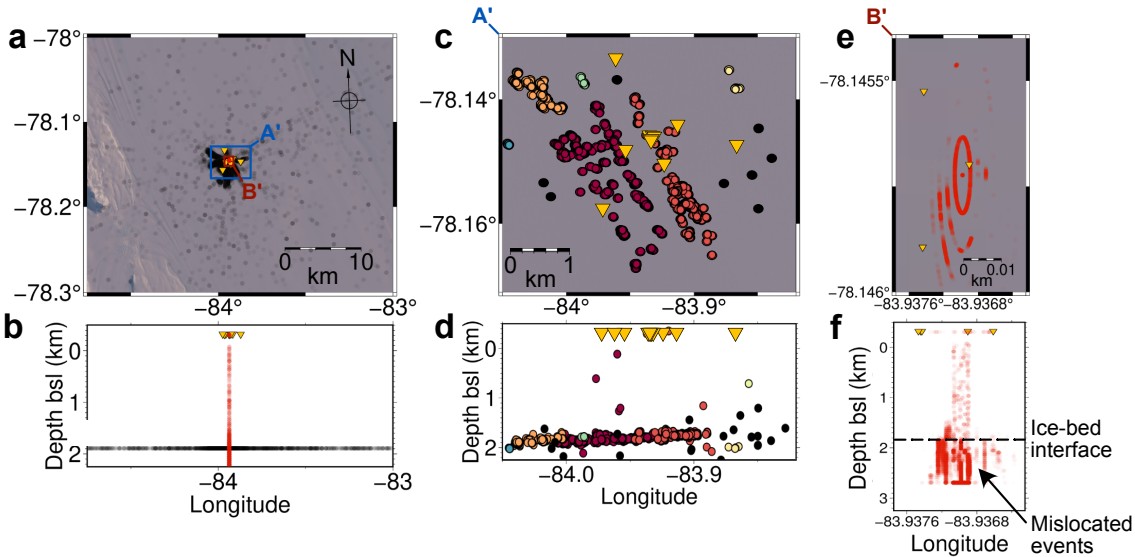

**Figure 5.** Comparison of array-derived to network-derived icequake catalogue. (a), (b) Array-derived icequake catalogue hypocentres. Red points are using the full 3D array location method. Grey points are using the fixed depth method (transparency corresponds to icequake amplitude). $A'$ and $B'$ denote the extent of the plotting in (c) and (e), respectively. (c), (d) Network-derived icequake catalogue hypocentres, coloured by icequake cluster (see text for details). (e), (f) Enlarged plot showing only the array-derived icequake hypocentres obtained using the full 3D array location method. As in (a), (b), transparency corresponds to icequake amplitude.

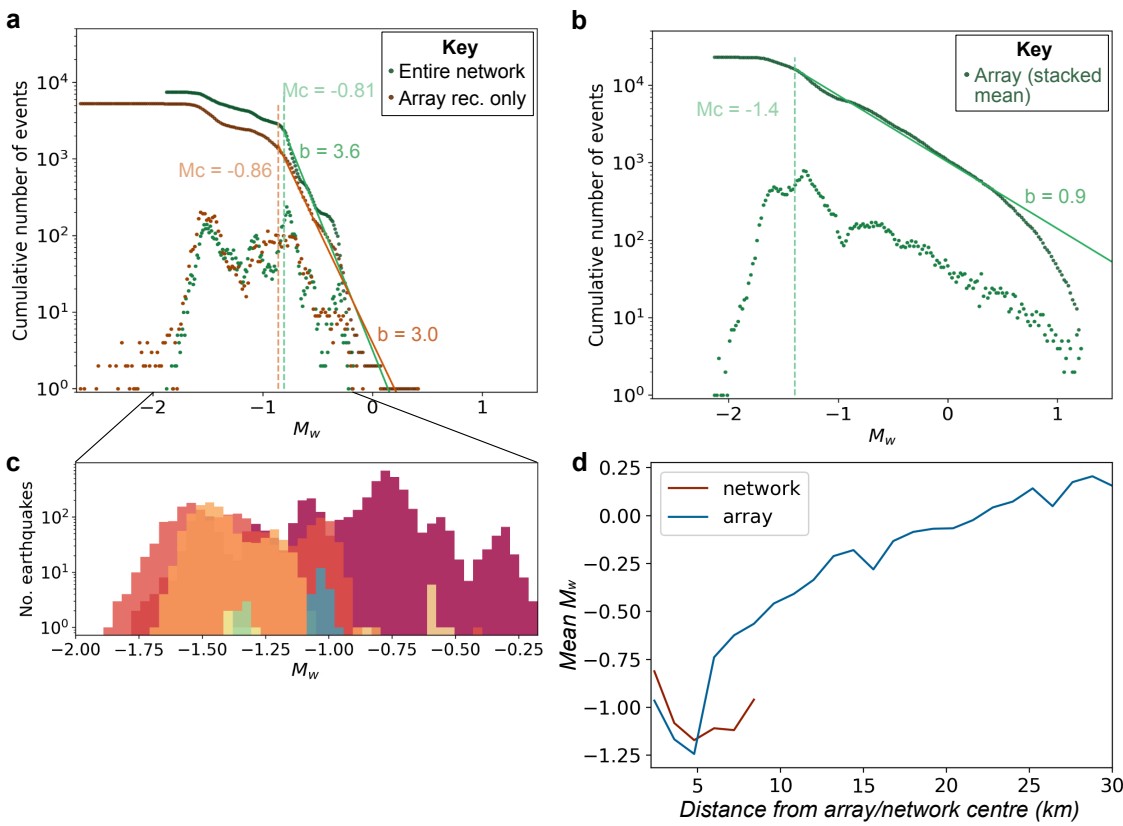

**Figure 6.** a. Moment magnitude distribution for entire network and array stations only, with events detected using migration-based method. Lower scatter points indicate data for each individual bin, while upper scatter points indicate cummulative moment magnitude distribution. b. Same as (a) but for array using array-based detection method (using the fixed-depth hypocentres, see Figure 5), and the mean of the linearly-stacked waveform data from the array. c. Plot of histogram-binned data for individual spatial icequake clusters from the entire network data from (a). Data are coloured to corresponding clusters in Figure 5c. d. Plot of mean $M_w$ with distance from the array/network centre for the entire network data from (a) and the array data from (b).

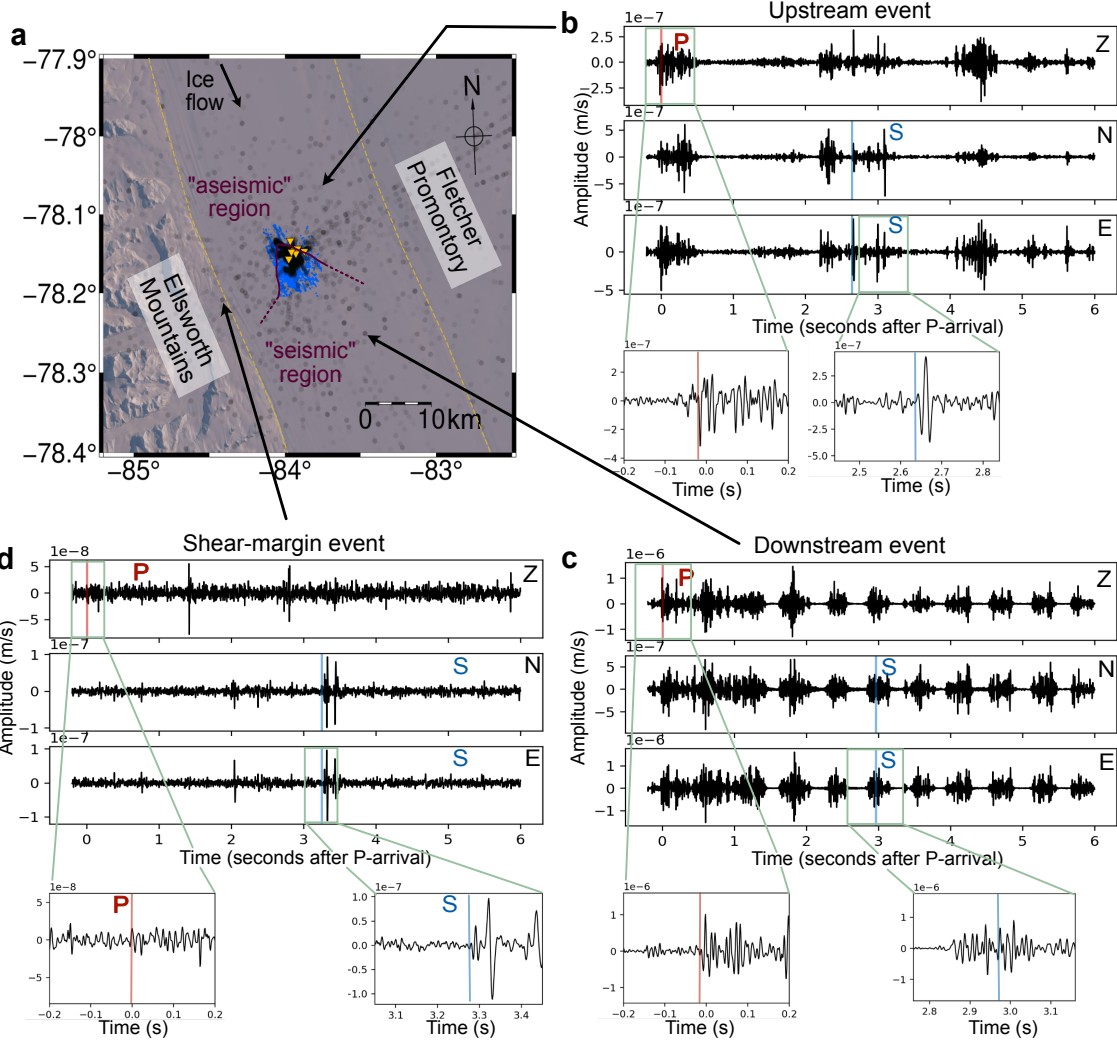

**Figure 7.** Examples of previously unobserved icequakes at Rutford Ice Stream. a. Map of overall seismicity detected in this study within the context of RIS more widely. Grey points are the icequake catalogue from array-processing in this study, blue points are icequakes from Kufner et al. (2021), the yellow dashed lines indicate the shear-margins and the purple solid-dashed line indicates the boundary between the previously inferred seismic-aseismic region (Smith, 1997; Smith and Murray, 2009). b. Example icequake located upstream, in the "aseismic" region from the $\Delta t_{P-S,max} = 2.5s$ icequake catalogue. c. Downstream icequake from a previously unstudied region. d. Shear-margin icequake, again from a previously unresolved region.