# Peer review of "Array processing in cryoseismology: A comparison to network-based approaches at an Antarctic ice stream"

_EGUsphere, 2023_

## Author Comment (AC1)

**Response to reviewer comments for the manuscript: "Array processing in cryoseismology: A comparison to network-based approaches at an Antarctic ice stream"**

We thank the reviewers for their valuable comments that have no doubt improved the manuscript. Below is a brief summary of the changes suggested and made, as well as detailed responses to the reviewers, stating exactly how we have taken into account their points. All our responses are in red.

Summary:
Both reviewers had comments broadly focussing on two aspects of the manuscript. The first points generally addressed the perception based on the original title that the manuscript acts as a review article, but that in reality the text did not form a review article, but rather a study using array processing. Our intention was never for the paper to act as a review article, and so we were very happy to adapt the title to address this point, as well as still adding a broader overview of the literature for interested readers. Hopefully these changes address the reviewers' first point. The second main point the reviewers' were concerned with regarded the array catalogue, specifically methods for identifying real events from false triggers. The reviewers' kindly suggested various ways to address this issue, in turn making the findings and conclusions more significant. To address this second point, we have now included both a slowness-ratio filter and a beam-power filter in order to try and minimise false triggers. This has proven particularly effective and we are very grateful for the reviewers steering us in this direction. It has definitely improved our confidence in our findings. Our focus of this work is to highlight the usefulness of array processing as a method for studying basal icequake seismicity at glaciers, in particular at large Antarctic Ice Streams. Therefore although the array-based icequake catalogues have changed somewhat from the original submission, the underlying methodology and findings hold true, with the changes made only acting to bolster the overall conclusions. The reviewers also suggested various minor formatting changes that we have also made. We thank the reviewers for all their contributions, which have no doubt improved the manuscript. We therefore are eager to resubmit an improved manuscript that will hopefully act as a useful basis and practical guide for further array-processing studies for glacial seismology, in particular basal icequake studies.

Reviewer 1:

In this study the authors apply seismic array processing to data recorded on an ice stream in Antarctica. The goal is to investigate the benefit of array processing (plane wave beamforming) compared to network processing for detecting and locating icequakes (migration and stacking). I do appreciate this study very much since I believe that array processing has a huge potential for analyzing complex and signals-rich cryoseismological data sets. A study comparing array and distributed network deployments on glaciers has not been done previously to my knowledge and is therefore very timely. It can be beneficial for researchers planning future field campaigns on glaciers where the number of sensors is limited, due to logistical reasons, to find the most optimal configuration for their research objectives. This is in contrast to recent large-N deployments on glacier, where multiple processing approaches can be tested after the measurements.

We thank the reviewer for their thorough review that has definitely improved the manuscript. Below we provide details of how we have addressed each of their points.

(1) One concern I have is not so much about the methods and results, but how this work is introduced and set in a broader context. The title seen alone suggests either a review article or the very first application of array processing for cryosphere monitoring. This is misleading since several studies have already been using arrays in this context for a while, and these works are not referred to in this manuscript. Array processing has been used on Alpine glaciers, in Greenland, in Svalbard and Antarctica. Please check the list of references below. Not all might be relevant and there might be even more. Some of these studies used permanent arrays (Antarctica; Svalbard). Others used temporary arrays deployments on or close to glaciers for the purpose of studying cryogenic seismic signals (icequakes, calving events, tremors, ...). Some do classic plane wave array processing; others apply matched field processing / general beamforming. The definition of what is array processing and what network processing in these studies may not be so clear always, however, I do believe these studies are related to your topic. Given these previous works and the sole focus on Antarctica in your study, I strongly suggest modifying the title to better reflect the content and contribution of this paper, and also update the text (mainly introduction) and references accordingly.

This is a good point. We wanted to aim to give an overview of array processing at glaciers and then present an implementation that allows us to compare between array processing and network based methods, specifically for microseismic studies. However, we have clearly not communicated this sufficiently well and so have changed the title, introductory and context text accordingly. The title is now: "Array processing in cryoseismology: A comparison to network-based approaches at an Antarctic ice stream". Hopefully the updated title and text now communicate this.

Thanks too for all the additional studies listed below. We were aware of a number of them, but not all of them. We initially didn't cite some as we didn't intend for the study to become a review paper. However, we are very much in favour of the final paragraph of the introduction providing as comprehensive a review of the literature as possible, so have added the majority of references to the text. Hopefully, although this section is not a comprehensive list that would be included in a review paper, it now acts to give the reader a more complete flavour of the applications of array methods in cryoseismology to date. Thanks again! Where we haven't included references, it is generally because they don't fall within our definition of array based methods. There are obviously some differences in how the community refers to arrays vs. networks. However, we clearly define where we draw the boundary/distinction in the first paragraph of the introduction. We are not suggesting that there is a right or wrong boundary to draw, but hopefully our mini-review paragraph is consistent with this definition.

Hopefully these changes address the concerns of the reviewer. We are very appreciative of this feedback, since it can be difficult as an author sometimes to put oneself into the perspective of an outside reader. Hopefully the scope of the work is now clear, with the focus being on advantages, challenges and the practical application of array-processing methods for icequake cryoseismology studies.

Examples of papers using array processing, beamforming, and FK analysis in cryoseismology:

https://doi.org/10.1785/0220200280

https://doi.org/10.1029/2021GL095996

https://doi.org/10.1017/aog.2018.25

https://doi.org/10.5194/tc-16-2527-2022

https://doi.org/10.14943/lowtemsci.75.15

https://doi.org/10.1029/2021GL097113

https://doi.org/10.5194/tc-14-1139-2020

https://doi.org/10.1002/2015GL064029

https://doi.org/10.1093/gji/ggac117

https://doi.org/10.5194/tc-13-3117-2019

https://doi.org/10.1002/2016GL070589

https://doi.org/10.3402/polar.v34.26178

(2) In Chapter 3.1.1 the array processing pipeline is introduced. Automatic array processing including phase detection by beamforming, phase classification by F-K analysis, and association using back-azimuth constraints, similar to your method, is routinely done at several national seismic data centers and at the IDC. This should be mentioned. See for example Chapter 9.9 in this one: https://doi.org/10.2312/GFZ.NMSOP_r1_ch9

Thanks, this is a good point. Although we wanted to be clear about the specific method we use, we now mention that similar methods are deployed at various seismic observatories globally, referencing that work (L76).

(3) You use beam power time series P_Z for detecting P waves and P_H for S waves. As far as I understand the slowness is not used to actually confirm that a P or S waves are detected, which is usually done in automatic array processing. Have you considered this? Theoretically, S waves can also be triggered by P_Z, and P waves by P_H, depending on the incident angle of the phase arrivals. Could it for example happen that a horizontally arriving P wave leads to a peak in P_H which you would miss because you are looking only for peaks in P_Z followed by P_H? I acknowledge that your procedure is more sensitive to detect deep icequakes, which is maybe what you intended. Also, there is of course the ray bending due to the firn layer. Some comments on that would be appreciated.

Good point. The reason we didn't use slowness initially for P and S wave identification is that the slowness we measure is actually apparent slowness, which is dependent on subsurface velocity structure. As we mention in the work, the firn layer causes significant issues since the velocity structure results in rays that dip steeply upward as they near the surface, which in turn significantly affect the apparent slowness observations. Indeed, this velocity structure is what motivated us to isolate P arrivals on the vertical and S waves on the horizontal, since a near-vertical arrival should isolate the various phases in this way. However, we think this is a great

point and have now implemented a slowness-ratio filter (Ss/Sp = vp / vs). The slowness-ratio rather than individual P and S slowness is useful because assuming the P and S waves have approximately the same ray path, then the apparent slowness component will cancel, and thus not affect the ratio. The slowness-ratio is used to remove potential false triggers, and appears to prove very effective at greater epicentral distances. We now describe this implementation in the methods, and its effects in the results/discussion. Thanks for the suggestion, it has definitely improved the array method.

(4) I am not sure about the motivation for Chapter 3.1.4. It is not clear where you do time-domain beamforming at this point. You described that you choose to do array processing in the frequency domain, i.e., F-K analysis in continuous data to measure the slowness vector, but nothing is mentioned about time-domain stacking in the detection and location procedure. Please clarify.

Sorry, this is obviously incorrect. When I was writing this I was thinking in the time-domain rather than the frequency-domain. We actually mean a phase-shift in the frequency domain. We have changed the text accordingly and are sorry for the confusion caused.

(5) I find it difficult to understand why your 3D location procedure fails so clearly, i.e., all events end up on a vertical line below the array. From the description of the methodology, I had the impression that the synthetic take-of angle and distance PDF should enable reasonable results. Could you add some sort of synthetic resolution test for the 3D method to investigate this? Your approach is by the way similar to how teleseismic events can be located using a single array. Using the 1D velocity models of the Earth, we can predict the horizontal slowness (ray parameter) of a P or S wave at a given distance. With the back-azimuth, we then have the location. Even if your firn model in not perfect, I expect more spread-out locations.

- Plot up column results in real detail – to show effect of firn layer

(6) Figure 6: Seeing these waveform examples makes me wonder if there could be many mis-associations in your detection list. Visually it is difficult to identify and relate the indicated P and S arrivals. I acknowledge that the authors have experience in identifying icequakes in that area. However, some proof for that these are not coincidental detections and associations would be appreciated. So how confident are you in the array-detected events?

- Address after new locations and phase associations…

Minor comments:

Chapter 1: I would already here emphasize that the set of 16 sensors is used for classic network processing later and add the dimension of it in the text, and that the set of 10 sensors used for array processing are located in the centre of the network. This would prepare the reader better for the comparison later. Side note: The 16 sensors together could also be considered as an array for processing longer wavelength signals, for example regional and teleseismic arrivals.

Good point. We've now added text making this clear in the introduction.

Line 4: Add calving as another source of cryoseismicity.

Good point. Added.

Line 100: You do you not explicitly write how you measure the take off angle. It is clear that it must come from the observed horizontal slowness and an assumed P/S wave velocity in ice, but good to mention this for the reader.

Good point. We now explicitly include how we calculate this in Equation 1.

Line 105: How do you obtain the path-averaged velocities?

From previous literature where they related seismic velocity to temperature. This is now described in the text, along with the values used (L138-140).

Caption figure 3:  I think this piece of text has to be deleted:  "n of an event alysis (FTAN)"

Sorry. Thanks for spotting that issue. It has now been removed.

Line 172: The comparison of network and array processing on the array is very interesting. But you could already mention here that the target signals are of course very different. For plane wave array processing you need events at some distance from the array, whereas the migration stacking is expected to work best for events inside the network or at close distance.

Good point to emphasise this point there in the text. Have now added text to communicate this in section 3.2. Thanks, definitely a clearer message now.

Line 181: Here it would be very helpful to distinguish between the two network processing approaches (all sensor and array only). I would assume that network processing outperforming the array processing for small magnitudes is mainly due to icequakes detected inside the dense array. This is expected since array processing needs plane waves as mentioned above. This discussion is missing and should be added.

We feel that although this is a valid point, the results and discussion dive into this detail/result in depth. We'd rather keep Section 3.3 as brief as possible and address this in Section 4 instead. Hopefully that is ok.

Chapter 4.1.1.: You give a good explanation for the peaks in the magnitude distribution. If my guess that the missing small events in the array processing results come mainly from events inside the array is correct, then these peaks could also originate from very close events which arrive with non-planar wavefronts. For evaluating this it would be very helpful to have the sensor location plotted in Fig 4c, and also show a zoom of array-detected events inside the network.

Interesting point. I think there might be a small misunderstanding here. With the array processing results, there are no significant missing small events, with the magnitude distribution being almost perfectly what one theoretically expects. The array results in Figure 5a,c are for the network-based detection and location method, and so can deal with non-planar wavefronts. To alleviate this confusion, we more clearly label Figure 5 on the plots themselves, so as to minimise this confusion. Apologies for that. We also now plot the receiver locations in Fig. 4c

and a zoom-in of the column directly beneath the network in Figure 4b. Hopefully that addresses all of the above comment. Thanks for drawing this to our attention.

- Add receivers to Figure 4c and provide an inset figure for 4b of the column detail.

Line 213: double "with"

Thanks. This is now corrected.

Chapter 4.2. As you write the comparison of the methods is difficult because not the same qc criteria are applied. One candidate for qc-ing array results would be the coherency or semblance for each detected phase. You could increase the threshold to see of the better constrained events resemble the network locations.

Yes, fair point. In theory it would be possible to create a coherency value that is comparable for both the network detections and the array-processing. However, this would be highly sensitive to the exact windows used (e.g. STA/LTA windows for the network-based approach and rolling window length for the beamforming). Although the windows used do obviously affect the detection of events, the exact values of the windows probably do not have much effect on whether a particular event is triggered, yet could affect the exact coherency values used for comparison. Noise levels would also be dependent on the exact parameters used. The shear volume of data processed ($10^5$ to $10^6$ events) would make calibrating detection parameters for each method to match the coherency a challenging task, which we deem not worth the gain. This is why we instead opt for using earthquake magnitudes (especially magnitude of completeness) to compare catalogues instead. Obviously this also has limitations, but does allow for the flexibility for others to test the two methods on other datasets more easily. We have also introduced the suggested false trigger filters for the array processing, which somewhat deal with this issue too (I.e. we have now effectively increased the detection threshold for the array-based detection using the slowness-ratio and beam-power filters, with local events to the array now resembling similar spatial trends to the network-based detections). Hopefully that addresses this point. We have endeavoured to adopt all the other suggestions of both reviewers, and hopefully this point has been sufficiently addressed to an acceptable level.

Fig 5: It looks like you show the cumulative and non-cumulative distributions, but the axes labels just says "cumulative".

The y-axis labels of Figure 5a,b are now modified to address this point.

- Change y-axis labels to state cumulative and non-cumulative.

Line 265: "… icequakes from" ?

Apologies for this grammatical error – we have now corrected it.

**Citation**: https://doi.org/10.5194/egusphere-2023-657-RC1

Reviewer 2:

In this article, Hudson et al. detect and locate glacial seismicity at Rutford Icestream (Antarctica) using both classical seismic networks and array-processing techniques. After explaining the methodology of both approaches including beamforming, waveform migration, triggering thresholds, etc, the authors compare the resulting event catalogs obtained from their deployment on Rutford Icestream. The main conclusion of this comparison is that the network-based approach detects less events compared to the array analysis. However, this advantage of arrays in terms of event number comes at the price that the individual event locations can be less good constrained, compared to using the network approach.

I do like the idea of applying beamforming to p- and s-wave arrivals separately and to subsequently combine the outcome to locate the events, because it does not require to detect and label the phases initially (the association is done based on the continuous beamforming results). This is a creative approach and an appropriate way to detect coherent arrivals close to or hidden in the background noise. Furthermore, the topic is well suited for the journal and provides an interesting discussion on deployment strategies in the harsh cryospheric environment. However, while the general idea is clear, I think there are some issues regarding the methodology including technical flaws and unclear formulations. For this reason, I am somewhat skeptical about the results and the conclusions drawn. Below, I provide major comments to address these concerns in detail, which are followed by a list of minor comments.

We thank the reviewer for their extensive comments that have no doubt improved the both the analysis and the manuscript, especially the suggestion regarding additional filters to discriminate between real events and false triggers. We have addressed all the reviewer's

comments, as described in detail below. Hopefully the improved manuscript is now in much better shape for publication.

**Major comments:**

Title: I think the title is a bit misleading, as it suggests a general paper on array processing in cryoseismology or a review paper. As the article focuses on detecting and locating high-frequency basal icequakes, I would suggest something like "Array processing to study icequakes on Rutford Ice Stream, Antarctica" or similar.

Fair point. The other reviewer also shared this view. We have therefore changed the title to: "Array processing in cryoseismology: A comparison to network-based approaches at an Antarctic ice stream".

Methodology section:

1) The authors state that the length of the time window used for f-k analysis is 0.025s, which is equivalent to 25 samples at 1000 Hz sampling rate. From this short window it will not be possible to get proper frequency spectra needed for the approach, in particular for the low frequencies used, i.e. down to 10 Hz. It is mentioned that this time window length is based on the "slowest time it would take for a seismic wave to travel across the array", which in my opinion must be 90 m / 1 km/s = 0.09 s, which is still hardly enough to properly resolve 10 Hz. I would appreciate if this is carefully checked and corrected.

There is indeed a trade-off between time window and frequency resolution. Our aim was generally to target signals at higher frequencies than 10 Hz, but would like to keep the frequency band as wide as possible, so as to detect events that have travelled further and are therefore more attenuated, with lower peak frequencies. To do this, we very much agree with the reviewer and so have reprocessed the entire dataset with a window length of 0.2 seconds. In order to still retrieve an adequate time resolution for phase arrival time picking, the windows now overlap, with the time step increment of 0.01 s. Thanks for raising this point. Although the previous parameters likely were still sensitive to local 'typical' icequakes with source frequencies ~60-80 Hz, our new values now are much more sensitive to events farther from the array. Hopefully this change now addresses the reviewer's concerns.

2) It is unclear to me, how the f-k analysis is conducted. Lines 74-78, which introduce the basic concept, are confusing in my opinion, e.g. why do you calculate power spectra for each receiver? What does point two mean - only that you calculate the beam power for the 2D slowness space? How do you calculate the results at discrete frequencies? As this is the core of this work, I believe it is necessary to clarify these points and I suggest to also add the formulas on which the f-k analysis is based.

Apologies for not adequately clarifying the exact beamforming method used. We have replaced the numbered list with continuous prose and equations detailing the entire beamforming calculation from power spectral densities for each receiver to beam power (see Eq. 1-4 and surrounding text). Hopefully this now clarifies the exact method used.

3) The proposed method distinguishes p- and s-waves through the beam power on the vertical and horizontal components, respectively. I would like to note here, that Sv waves may map into the vertical component and p-waves that come from outside the array may map into the horizontal components (though it is noted that due to the firn layer there is a steep incident angle). In addition, Rayleigh waves will show up on both vertical and horizontal components. I think it would thus be helpful to take the p-to-s velocity ratio of an icequake detection as an additional criterion to trigger the location process. There should be a clear difference in p- and s-velocity reflecting the subsurface p-to-s ratio. In any case, I think the manuscript should also contain a time series of the slowness and back azimuth associated with the maximum beam power shown in Figure 3. This would also help the reader to better understand the basic concept.

This is a good point. We have added a brief note in the text about our assumption that all P and S wave energy is on the vertical and horizontal components, respectively. We also very much like the idea of the vp/vs ratio criterion and so have implemented it (using the slowness ratio). It works really well, and has significantly improved the event catalogue by reducing the number of false triggers (and likely surface wave triggers). Hopefully the addition is now sufficiently described in the text (see L160-166). Thanks for pointing this out – it has significantly improved our analysis.

We have also plotted slowness and back-azimuth in Figure 3.

4) Connected to the previous comment: the f-k method yields slowness/velocity values and large parts of the location analysis requires a velocity model. However, the manuscript does not report any values of seismic velocity, neither in the context of the beamforming results nor in the context of the velocity model. I strongly encourage the authors to include more detail to remove ambiguity.

Good point. We now include information in the text describing the velocity model (Section 3.1.2), as well as now including it in the Zenodo data repository (https://doi.org/10.5281/zenodo.8120941 ).

Results and conclusions

Based on various aspects, I feel that the conclusions of the paper are drawn on somewhat little evidence. The authors admit (which is not a problem per se), that explicit 3D locations of icequakes using the array method are not possible (all events locate on a vertical line beneath the array, Fig. 4b). For this reason, the proposed method prescribes a fixed depth for all events assuming basal seismicity, only. However, repeated surface icequakes producing dominant Rayleigh waves (which can be the dominant event type on glaciers) would be picked up in the array analysis as well (as Rayleigh waves will peak on both vertical and horizontal beam power time series). The authors state, that false triggers are contained in the catalogue, e.g. such surface crevasse fracture events. For these events, the location method based on p-s arrival times and basal seismicity is not justified (or actually violated), which may lead to erroneous event locations and interpretation of the seismicity. Finally, in the presented examples of newly detected events using the array (Figs. 6c and 6d) I cannot discern obvious arrivals, which strengthens my view that the catalogue may contain many false triggers. In total, due to the various issues outlined above, I feel that better evidence needs to be presented to support the

conclusions. I think that this may be possible by choosing a more strict trigger criteria/thresholds.

Hopefully now we have made the reviewer's other suggested changes, the results now allow us to strengthen the conclusions. We now strengthen the conclusions by mentioning the slowness filter and how it allows us to discriminate between distal icequakes and false triggers better. This then allows us to more confidently conclude that we can use arrays to detect icequakes over a greater spatial footprint than networks, as well as otherwise inaccessible areas of ice streams. This will in turn allow for more comprehensive cryoseismology studies, at least of basal processes.

**Other comments:**

Line 7 (abstract): I think here it is necessary to specify that the method is tailored to high-frequency (>10 Hz) events with distinct p- and s-wave arrivals. For events with dominant Rayleigh waves (which are very numerous in on-ice recordings), the method will not work, as the p-s criterion breaks down and as there will be a peak in beam power on both the vertical and horizontal components.

- Maybe have a minimum P-S time, in order to discriminate between P,S waves and Rayleigh waves

Line 9 (abstract): Also here, "cryosphere applications" is too general in my opinion (see above).

Fair point. We have changed the text to: "body-wave cryoseismology applications".

Line 22-23: I guess with "such icequakes" you mean basal seismicity? Because especially for surface crevassing events, there are several studies that use arrays, e.g. Mikesell et al. (2012), Köhler et al. (2016), Lindner et al. (2019).

Yes, sorry. We have now clarified this by substituting "such" for "basal".

Lines 24 and 26: I am having trouble with the term "sensitive", as this implies that a network would not sense any events outside the network and an array would not sense events within the array, which of course is not true. So maybe "designed" would be more appropriate here?

Good point. We originally included "primarily only sensitive" but have changed the text to "predominantly sensitive" for each method, since each method is dominantly sensitive to either events inside or outside the aperture of the network/array. Hopefully that is now acceptable. It is a key point of the whole piece of work, so we'd like to keep this point in some form.

Line 28: Is it really justified to generally say that arrays enable event detection at greater distances? In the end, a network is a large array of sensors as well.

We think it is justified, since by our stated definition of arrays, they are sensitive predominantly to events outside the array. Indeed, this is why arrays are typically used for nuclear test ban treaty monitoring, for example. We have now added this to the text, to communicate this point more clearly (L29-30).

Line 35: Problem with reference rendering

Thanks for spotting this. It is now sorted.

Line 37: I think there is a problem with the references: only the paper by Klaasen et al. looked into icequakes, but not the other two papers. Please check.

Sorry, this is not a problem with the referencing, but rather our wording. We meant to state "detection of seismicity" generally, not "icequake detection", as was originally stated. We have now updated the text accordingly.

Line 38 and following: Actually, there are a number of other examples of array analysis and beamforming to locate icequeakes, e.g. Mikesell et al. (2012), Köhler et al. (2016), Lindner et al. (2019).

We realise that we haven't been specific with our definition of icequakes that we are interested in. We are explicitly talking about basal icequakes, and so have now clarified this in the text. We have added the Lindner reference anyway, as that is indeed a good example of using arrays to detect and locate surface crevassing seismicity using beamforming. We have also added the Kohler reference, but again it is to locate calving and crevassing seismicity rather than basal icequakes. The Mikesell paper is not cited as they don't use array-based/beamforming methods for their detection or location. Thanks for bringing this point up though, as we would have regretted not bringing these pieces of work to the attention of readers, as they are excellent studies that are somewhat related to our work.

Lines 47-51: Can you provide a few more details on sensor installations and deployment length? Also, it would be nice to get some more context on the study site.

Fair point. We've added a note in Section 2 to state that they are buried and the exact dates they were deployed for. Otherwise, the sampling rate, instrument type and array geometry are already specified.

Lines 93-96: I do no understand the third criterion, can you please rephrase it?

Apologies for this. It has now been rephrased.

Line 106: Maybe add here that the fixed depth method assumes that the events originate at the glacier bed.

Good point. Added a sentence to state this.

Line 107 and following lines: I cannot follow, how the 3D location method works. Also I am afraid that figure 2e does not help to get the concept. I suggest to rewrite this paragraph.

Apologies for any poor communication of the method. We have now rewritten that paragraph to hopefully better communicate the concept.

Line 112: Which velocity model are you using here and how does it look like? I think this is important to mention as you later say you "assume that there is negligible uncertainty in the velocity model."

Apologies. We now include a section explicitly describing the velocity model in detail (Section 3.1.2).

Line 134: Why "approximate" seismic velocity?

Thanks for pointing that out. It is unnecessary/confusing and we have therefore removed the word "approximate".

Line 135 and following lines: The provided frequency limits assume slightly different velocity values for the min and max receiver spacings. Please clarify and mention which velocities are used.

Apologies, but we don't quite follow this point. We state the receiver spacing and therefore the frequency sensitivity for P and S waves is based on the velocity for P and S waves. For clarification, we have now added though that theses values are using the bulk ice P and S waves, respectively. Hopefully that clarifies any issues.

Line 140: This sentence implies that you do not use a beamforming technique in the frequency domain? In an earlier part of the manuscript, you refer to this paragraph for the stacking of the beamforming maps in the 2D slowness domain, but here waveforms are mentioned. What are you stacking here, please specify.

Good point. The other reviewer also raised this point and it was a slip in our wording, apologies. In the text, we have changed "time-shifted" to "phase-shifted". Hopefully the new text clarifies that we are indeed performing beamforming in the frequency domain.

Line 148 and following lines: I guess also here, the velocity model is crucial.

Sure, good point. We now have a specific subsection detailing the velocity model (Section 3.1.2).

Lines 153-155: Why do you need to relocate the events from the QuakeMigrate method, please specify.

Apologies for not clarifying this. QuakeMigrate has been deliberately designed to locate events on the nearest grid cell within the 3D search space, for computational efficiency. The locations are therefore approximate and not based on probabilistic methods. NonLinLoc is a probabilistic relocation software that finds the optimal location without such constraint, and because it is probabilistic it provides a better estimate of uncertainty with which one can filter the data by physically meaningful parameters. Thanks for raising the point – we now succinctly elaborate on the above in the text.

Line 178: Why is there a sharp detection limit at ~7.5 km for the network?

Sorry for not adequately describing why before. We have now added the text at that point: "This limit is the consequence of the boundary of the network-based search grid, with the

search area significantly exceeding the physical microseismic detection limits of the network-based algorithm." I hope the study doesn't come across as comparing apples and oranges. However, there is simply no point running search grids of greater sizes for the network-based method, since no events are detected at such distances and the computational cost of increasing the search grid is high (we already run the method on a HPC using significant computational resources).

Lines 183-185: Does that mean that the results of the array-based method contain false-triggers?

Indeed it does. Your comments on possible slowness filters have now actually addressed this point to some extent, but one of the challenges of the array-based method compared to the network-based method is the limited number of parameters that can be used to filter false triggers. Just for the reviewer's interest: When developing QuakeMigrate, we spent a long time developing physically-meaningful parameters that could be used to filter the data. I think now with the reviewer's additional filter suggestions, we now have a more robust false-trigger filter than previously (thanks!) but it does still likely not perform as well as QuakeMigrate's false-trigger filters.

Line 213: Twice "within" at the end of the line.

Apologies for not spotting that – don't know how we missed it. Removed the second "within".

**FIGURES**

Figure 1: Most of the subfigures are too small, it is therefore not possible to get the geographical context of the field site. Please make bigger.

We've made the subfigures bigger, including the text. Hopefully the subfigures are now of adequate size.

Figure 2: Again many fonts are too small. In panel c: numbers are missing on the slowness axis. Maybe also use "beam power" instead of "power" only (throughout the manuscript) to avoid misinterpretation?

We have made the relevant changes to the figure and changed "power" to "beam-power" throughout the manuscript. Hopefully that now makes things clearer in the manuscript.

Figure 3:

As suggested earlier, it would be great to see also the slowness and back azimuth time series together with the beam power.

We've added those data as subplots, with the labelled phase arrivals. Hopefully that improves the figure.

Where do the focal mechanisms come from?

These are schematic only, they are not actual inversions. We have now clarified that in the figure caption and in the main text.

Legend: What is "FTAN"? "events.n" > events?

I think these are now removed.

Figure 4: What are the different gray scales of event locations in panel a? Panels c and d: I suggest to also show the locations of the geophones here.

The grey scales are relative amplitude of the icequakes (level of transparency/opacity corresponds to amplitude). We now clarify this in the figure caption. In Figure c,d we have now added the geophone locations.

Figure 5: This figure is again too small. I am also having trouble to understand panels a and b: maybe I am misunderstanding something here, but why are there two values of the cumulative event number for a given magnitude? E.g. in panel b, for Mw -0.5, the cumulative event number is both ~3*10**3 and ~6*10**4?

The figure has now been enlarged. We have also clarified in the caption that the lower scatter data are for individual bins and the upper data are the cumulative values. Hopefully that now makes sense. This is a common way to plot magnitude distributions, so that one can see any bumps in the individual bin data (which we discuss in the text), that would otherwise be mostly hidden in the cumulative distribution.

Figure 6: The subpanels showing phase detection are too small.

I think we have now addressed this. There is ultimately quite a lot of information that we want to convey in this figure, so hopefully we have communicated it adequately now while it is sufficiently clear.

Legend: "reion" > region

Good spot – made the change.

References:

Köhler, A., Nuth, C., Kohler, J., Berthier, E., Weidle, C., & Schweitzer, J. (2016). A 15 year record of frontal glacier ablation rates estimated from seismic data. Geophysical Research Letters, 43(23), 12-155.

Lindner, F., Laske, G., Walter, F., & Doran, A. K. (2019). Crevasse-induced Rayleigh-wave azimuthal anisotropy on Glacier de la Plaine Morte, Switzerland. Annals of Glaciology, 60(79), 96-111.

Mikesell, T. D., van Wijk, K., Haney, M. M., Bradford, J. H., Marshall, H. P., & Harper, J. T. (2012). Monitoring glacier surface seismicity in time and space using Rayleigh waves. Journal of Geophysical Research: Earth Surface, 117(F2).

**Citation**: https://doi.org/10.5194/egusphere-2023-657-RC2

---

## Author Response (AR3)

**Response to reviewer comments for the manuscript: "Array processing in cryoseismology: A comparison to network-based approaches at an Antarctic ice stream"**

We thank the reviewer very much for their additional valuable comments that have further improved the manuscript. Below are as detailed responses to the reviewers, stating exactly how we have addressed their points. All our responses are in red.

**Reviewer comments:**

The authors addressed most of my comments adequately and strengthened the results by repossessing their data using additional filters. Overall, this manuscript has improved sufficiently and can be published after addressing a few issues:

It is not 100% clear to me how the PDFs are used to locate the icequakes. How does the measured takeoff angle and S-P difference are related to the pre-computed values for each location? As far as I understand the PDF represents the probability of a source location given angle and S-P. Is the probability simply the misfit of observed and theoretical values for a given source location?

The overall PDF represents a stack of: the probability of an event arrival with a given take-off angle (both for P and S); and an event arrival having the event S-P time. Individual PDFs do indeed comprise of the misfit between observed and theoretical values. We now communicate this more clearly in the text (L151-153). Hopefully it is now clear. Apologies for the miscommunication/lack of clarity previously.

Figure 5: Now it is clear that the array-based and 3D-located icequakes are not on a vertical line. So, the icequake locations from both location methods are completely different. Or maybe I miss something? You write:" Uncertainty in the velocity structure of the firn layer, especially at P-wave wavelengths (< 10 m) limits the measurement of takeoff angle from apparent slowness used in the array-based method's 3D icequake location procedure. This is what causes the icequakes located using the array-based method to be miss-located directly beneath the array (red scatter points, Figure 5a,b)." Does this mean that the 3D location method fails completely? Wouldn't it be best then to remove this from the paper? You probably have good reason why you want to keep it.

Indeed. That is why we don't use the 3D method further. We include it to show how it fails, caused by the velocity structure of the firn layer take-off angles, in order to emphasise why it is important to account for near-surface velocity structure when performing array processing. We debated whether or not to include it, and decided to include it because it could be useful for deployments with no firn-layer (i.e. directly on ice). We have now endeavoured to make this even clearer in the text (L305-320) and extended the axis of Figure 5f to really exemplify the issue (labelling it on the figure too).

(By the way, it seems that your response to that point in the response letter is incomplete: "• Plot up column results in real detail – to show effect of firn layer")

Figure 7: ?? in caption. For an analyst it would be very difficult to visually associate these arrivals. I suppose the slowness filter ratio and the requirement of similar back-azimuth makes the association unambiguous for these examples (?). Plotting these examples on the more distant network stations would probably make it easier to verify that these phases belong together based on the move-out.

(also here the response is incomplete: "• Address after new locations and phase associations…")
Apologies for this. We made the change, so not sure why we didn't remove the comment.

Line 160: delete «We also create»
Thanks. Done.

Line 321: space missing "icequakeare"
Thanks. Change made.

---

## Author Response (AR4)

**Response to editor comments for the manuscript: "Array processing in cryoseismology: A comparison to network-based approaches at an Antarctic ice stream"**

We thank the editor for their additional points that improve the clarity of the manuscript and address some typological errors and data availability comments. Below is a detailed response to the editor's comments, stating exactly how we have addressed their points. All our responses are in red.

**Editor comments:**

[Fig. 7]

Both, a reply and correction in regard to the following remark by the Reviewer are missing.

Apologies for this. Please see the response we should have included previously.

"Figure 7: ?? in caption. For an analyst it would be very difficult to visually associate these arrivals. I suppose the slowness filter ratio and the requirement of similar back-azimuth makes the association unambiguous for these examples (?). Plotting these examples on the more distant network stations would probably make it easier to verify that these phases belong together based on the move-out."

We have dealt with the reference issue that caused the ??. Indeed, the slowness filter ratio and back-azimuth requirement make the association as unambiguous as possible, although there is obviously still a chance that phases are mis-associated. We are really pushing the capability of the technique here, and that is hopefully communicated in the text. We choose not to plot the moveout as the events are typically far from all the stations, not just the inner array stations, so would all look very close together and therefore not be very diagnostic. Hopefully that is ok.

Moreover, please re-check the necessity of 2 arrows. Secondary lines are pointing from subplots (d; short) & (c; long) toward (a), which look redundant to me.

Good point – sorry for this. We have now removed all superfluous arrows.

[Line 398: "DOI tbc"]

I am sorry to be picky, but according to our data policy, the data should be openly available and provided with a DOI in the reference list. I have seen many data statements like this and, on several occasions, have never heard back from the authors after asking the whereabouts of their data.

Quite right to be picky. Its really important that data is openly accessible. Apologies that we didn't update this. All the data is now on IRIS and the correct DOI is provided in these statements now.

[Line 97]

n((u
- the second parenthesis seems to be in bold for no reason.

The () here around the **u_i** are not needed, but the bold type is needed, as this parameter is a vector (of length 3). See on the RHS of the equation that we have u_x,i, u_y,i. We have therefore removed the () but retain the bold type. Hope that makes sense.